# Impact of Nystatin Oral Rinse on Salivary and Supragingival Microbial Community among Adults with Oral Candidiasis

**DOI:** 10.3390/microorganisms11061497

**Published:** 2023-06-05

**Authors:** Lanxin Zhang, Samantha Manning, Tong Tong Wu, Yan Zeng, Aaron Lee, Yan Wu, Bruce J. Paster, George Chen, Kevin Fiscella, Jin Xiao

**Affiliations:** 1Department of Molecular and Cell Biology, University of California Berkeley, Berkeley, CA 94720, USA; zhanglanxin125@berkeley.edu; 2Department of Biostatistics and Computational Biology, University of Rochester Medical Center, Rochester, NY 14627, USA; samantha_manning@urmc.rochester.edu (S.M.); tongtong_wu@urmc.rochester.edu (T.T.W.); 3Eastman Institute for Oral Health, University of Rochester Medical Center, Rochester, NY 14627, USA; yan_zeng@urmc.rochester.edu (Y.Z.); aaron_lee@urmc.rochester.edu (A.L.); yan_wu@urmc.rochester.edu (Y.W.); 4Department of Microbiology, Forsyth Institute, Cambridge, MA 02142, USA; bpaster@forsyth.org (B.J.P.); tchen@forsyth.org (G.C.); 5Department of Family Medicine, University of Rochester Medical Center, Rochester, NY 14627, USA; kevin_fiscella@urmc.rochester.edu

**Keywords:** Nystatin, oral microbial analysis, *Prevotella*, *Streptococcus*, *Veillonella*, *Actinomyces*

## Abstract

This study aimed to evaluate the impact of Nystatin oral rinse on salivary and supragingival microbiota in adults with oral candidiasis and identify predictive factors related to individuals’ responses to Nystatin. The trial involved twenty participants who used 600,000 International Units/application of Nystatin oral rinse for seven days, four times a day, and were followed up at one week and three months after the rinse. The salivary and plaque microbiome of the participants were assessed via 16S rDNA amplicon sequencing. Overall, salivary and plaque microbiomes remained stable. However, among the participants (53 percent) who responded to Nystatin rinse (defined as free of oral *Candida albicans* post treatment), *Veillonella* emerged as a core genus alongside *Streptococcus* and *Actinomyces* in supragingival plaque at the 3-month follow-up. Furthermore, statistical models were fit to identify predictive factors of Nystatin rinse success (elimination of *C. albicans*) or failure (remaining *C. albicans*). The results revealed that an increased level of salivary Interferon (IFN)-γ-inducible protein (IP-10), also known as C-X-C motif chemokine ligand 10 (CXCL10), was an indicator of a failure of responding to Nystatin rinse. Future clinical trials are warranted to comprehensively assess the impact of antifungal treatment on the oral flora.

## 1. Introduction

Dental caries and periodontal diseases are common infectious oral pathologies where an altered ecology of the oral microbiota and its intricate microbial network could play an important etiological role [1,2,3]. Recent efforts to understand the oral microbiome have shed light on its fungal component [4]. In particular, *Candida* thrives in the presence of lower oral pH and is enriched in caries, with mechanistic studies suggesting it participates in the disease process by synergistically interacting with acidogenic bacteria [1]. Increased prevalence and abundance of *Candida* were reported associated with advanced caries [5]. 

The role of *Candida albicans* (*C. albicans*) in children with early childhood caries (ECC) was more definitive. A comparison of the pooled plaque mycobiome of 40 children with ECC to the same number of caries-free children found that the genus *Candida* was enriched in ECC [6]. O’Connell et al. [7] (2020) further showed a trend for decreased mycobiome diversity as caries severity increased and found *C. albicans* and *Candida dubliniensis* (*C. dubliniensis*) were positively correlated with caries. Consistent with the point that *C. albicans* is associated with severe ECC, a meta-analysis [8] indicated that children with oral *C. albicans* had >5 times higher odds of having ECC compared with those without *C. albicans* and children with early-life oral *Candida* colonization were at higher risk for *Streptococcus mutants* (*S. mutans*) emergence by 1 year of age [9].

The periodontal pocket may also be a niche for the existence of *Candida*, but whether this fact is the cause or consequence of periodontal disease remains unclear. A meta-analysis [10] demonstrated that *Candida* spp. detection rate and density were statistically significantly higher in patients with chronic periodontitis than in subjects with clinically healthy periodontium. In vitro studies [11,12] further characterized the interaction between *C. albicans* and subgingival bacterial plaque constituents such as *Fusobacterium nucleatum* (*F. nucleatum*) and *Porphyromonas gingivalis* (*P. gingivalis*), which might serve to increase periodontal dysbiosis. However, a study in an animal model [13] provided discordant evidence that *C. albicans* shielded the periodontal killer *P. gingivalis* from recognition by the host’s immune system. This study [13] showed co-infection of *P. gingivalis* and *C. albicans* led to milder inflammation. It is plausible to assume that the mixed biofilm would be more resistant to antibiotic treatment.

Therefore, adjuvant treatment with antifungal agents may be beneficial for prevention or treatment of caries or periodontal diseases [14]. The most commonly used antifungal medications for oral candidiasis are Nystatin oral rinse. Nystatin is a membrane-active polyene macrolide produced by *Streptomyces noursei* strains and is available in various forms, such as oral suspension [15]. Our recent in vitro study [16] showed that Nystatin application led to a reduced formation of *C. albicans* and *S. mutans* dual-species biofilms and cariogenic microcolonies. Furthermore, our previous clinical trial [17] indicated that Nystatin oral rinse reduced *S. mutans* salivary carriage among adults, despite only 53% of the participants responding to Nystatin rinse (defined as free of oral *C. albicans*). The goal of this study was to (a) assess the impact of Nystatin oral rinse on salivary and supragingival microbiota among healthy adults, (b) identify factors (e.g., oral microbiome and immune markers) that could predict a host’s response to Nystatin antifungal treatment.

## 2. Materials and Methods

### 2.1. Study Population

#### 2.1.1. Study Design

We conducted a clinical trial on twenty participants who met the required criteria for inclusion and exclusion, at the Eastman Institute for Oral Health (EIOH), University of Rochester Medical Center (URMC). The trial was a single-arm and non-randomized one, and its protocol received approval from the University of Rochester Research Subject Review Board (#STUDY00004638). The study was registered on Trials.gov (#NCT04550546) and followed the Strengthening the Reporting of Observational Studies in Epidemiology (STROBE) reporting guideline.

#### 2.1.2. Participants

The study participants were recruited from two sources, namely the existing pool of patients at URMC-EIOH clinics, and the volunteers from the general community in Rochester NY.

To be eligible for the study, the participants needed to meet certain inclusion criteria, including being 18 years or older, having a positive oral *Candida* detection with sufficient oral *Candida* burden (≥400 Colony Forming Unit (CFU)/mL of salivary *Candida*) that could be diagnosed with oral candidiasis using the laboratory standard established by Epstein et al. [18]. Eligible participants also needed to have at least 10,000 CFU/mL of *S. mutans* in saliva.

The exclusion criteria included having visible signs of candidiasis on the mucosa or tongue at screening, having systemic diseases such as HIV, cancer or diabetes, having used local (oral) or systemic antibiotics or antifungal medication within the last three months, being pregnant or breastfeeding, having more than eight missing teeth (excluding third molars and orthodontically extracted teeth), having more than four decayed teeth, having a removable dental prosthesis used to restore missing teeth, or being allergic to Nystatin. A pregnancy test (urine test) was conducted for verification.

#### 2.1.3. Study Procedures

During the trial, participants rinsed their mouths with 6 mL of Nystatin suspension (100,000 IU/mL) four times a day (every six hours) for one week. The investigators emphasized that the suspension should not be swallowed. The URMC Investigational Drug Services provided the Nystatin oral suspension. The participants returned the Nystatin suspension bottle so that the remaining quantity could be measured to assess their adherence to the treatment. Additionally, the study participants filled out a treatment-adherence log. The amount of medication usage among participants was 24.4 ± 4.8 doses. Only one participant complained of mild side effects that cleared up shortly (nausea and sore throat) after taking the third dose on the third day.

#### 2.1.4. Study Flow 

The study participants were assessed at three time points: Baseline visit (V1); 1 week after the completion of Nystatin oral rinse (V2); and 3 months after the completion of Nystatin oral rinse (V3).

#### 2.1.5. Data Collection and Examination 

The trial procedure has been detailed previously [17]. Briefly, during the baseline visit, a questionnaire was used to collect data on the participants’ demographic and socioeconomic background, and oral hygiene practice. Data on medical history and medications were self-reported by the participants and verified using electronic medical records. The medical background information collected included physician-diagnosed systemic diseases such as hypertension, diabetes, asthma, anxiety, depression and kidney disease, etc. 

A comprehensive examination was conducted at the baseline and each subsequent study visit was managed by a dentist in a dedicated examination room at URMC, using standard dental examination equipment, materials and supplies. Caries was scored using the DMFT (decayed, missing and filled teeth) index. To assess gingival inflammation, bleeding on probing (BOP) was evaluated. Dental plaque accumulation was assessed using the Plaque Index (PI), as described by Löe [19]. Each of the four gingival areas of the tooth was given a score ranging from 0 to 3.

#### 2.1.6. Saliva and Plaque Sample Collection and Processing

Methods used for saliva and plaque sample collection have been detailed previously [20,21]. The researchers collected approximately 2 mL of non-stimulated saliva samples by asking the study participants to spit into a sterile 50 mL centrifuge tube. Prior to the sample collection, the study participants were instructed not to eat, drink or brush their teeth for 2 h. Supragingival plaques from the whole dentition were collected using a sterilized periodontal scaler. The plaque samples were suspended in 1 mL of a 0.9% sodium chloride solution in a sterilized Eppendorf tube. The clinical samples (saliva and plaque) were kept on ice and transported to the laboratory within 2 h for further testing. BBL™ CHROMagar™ *Candida* was used to isolate and identify *C. albicans* based on the colony color. Colony PCR was used for a further identification of those *Candida* spp. that could not be identified via colony morphology [9]. Salivary cytokine/chemokine at three time points (baseline, 1 week and 3 months after Nystatin rinse) was assessed in the University of Rochester Human Immunology Center Core Lab facility, detailed previously [17]. Immune marker data from the trial were used for statistical analysis.

### 2.2. DNA Extraction and 16S rDNA Sequencing

DNA was extracted from 200 uL of saliva samples and supragingival suspension using the ZymoBIOMICS^®^ DNA Miniprep Kit at the Forsyth Institute, Cambridge, MA, USA. The Quick-16S™ NGS Library Prep Kit was used to prepare the DNA samples for targeted sequencing, with custom-designed primers from Zymo Research that provided optimal coverage of the 16S gene while maintaining high sensitivity. The primer set used was Quick-16S™ Primer Set V1–V3. The library was prepared using real-time PCR to control cycles and minimize PCR chimera formation. The PCR products were quantified with qPCR fluorescence readings and pooled together based on equal molarity. The final pooled library was cleaned up and quantified using TapeStation^®^ and Qubit^®^. Positive controls, such as the ZymoBIOMICS^®^ Microbial Community Standard for DNA extraction and the ZymoBIOMICS^®^ Microbial Community DNA Standard for targeted library preparation, were used, along with negative controls (i.e., blank extraction control, blank library preparation control), to assess bioburden levels. The final library was sequenced on Illumina^®^ MiSeq™ with a V3 reagent kit (600 cycles, San Diego, CA, USA), with 10% PhiX spike-in.

### 2.3. Microbiome and Statistical Analysis

QIIME 2 [22] was used to quantify the composition and diversity of each community based on its open-reference OTU picking facility. Only sequencing data that met quality control standards were used to create a caries prediction model in this study. These data were divided into operational taxonomic units (OTUs), and only those OTUs with features containing at least four counts in at least 20% of the data and with at least 10% variance (measured by inter-quantile range) were included in downstream analysis.

The samples were grouped into four groups: plaque samples collected from individuals who responded or did not respond to Nystatin rinse and saliva samples from the subjects who responded or did not respond. Each group contains samples from three visits: the baseline, 2 weeks and 3 months. The changes in the oral microbial composition of the samples were assessed.

The study assessed alpha diversity and beta diversity measures to analyze the diversity of the microbiota. The Shannon index was used for alpha diversity, and non-phylogenetic Bray–Curtis distance was used for beta diversity. PCoA was used to visualize the beta diversity results in a 2D plot. Additionally, relative abundance plots were created to visualize differences in microbial composition between visits. T-tests for correlated samples and PERMANOVA were used to evaluate the statistical significance of the results, with a significance level set to 0.05.

The ANCOM-BC model [23] was applied to provide confidence intervals for differentially abundant taxa. It addresses the problem introduced by differences in the sampling fractions across samples. Linear discriminant analysis (LDA) coupled with effect size (LEfSe) analysis [24] and non-parametric factorial Kruskal–Wallis sum-rank test were used to identify differentially abundant features between the baseline and 1-week follow-up in dental plaque. 

The Core microbiome analysis was performed using MicrobiomeAnalyst [25,26]. The results were shown as heatmaps containing taxa detected in over 20 percent of the population and with certain relative abundances. The *Y*-axis represents the prevalence of the taxa given the detection threshold (relative abundance (%)) on the *X*-axis. Heatmaps were employed to compare the core taxa of the samples collected at the baseline, 2 weeks and 3 months. 

Pearson’s correlation analysis was performed to calculate pairwise correlations between taxa at the genus level among dental plaque whose participants responded to Nystatin oral rinse. The correlation filter was set up as >0.6 and *p* < 0.05. The correlation results identify potential interactions between microbes that could represent mutualistic, commensal, or competitive relationships. The pie chart indicates the mean of taxa-relative abundance at the related study visit.

### 2.4. Classification Analysis

Furthermore, four statistical models were fit with the intention of identifying predictive factors of Nystatin rinse success or failure. The first model was a generalized linear mixture model with a least absolute shrinkage and selection operator (LASSO) penalty to perform variable selection. This model took into account whether or not the subject was *Candida*-positive in saliva at each visit (all yes at baseline). It used demographic characteristics, medical background, salivary immune marker (measured and reported previously [17]), oral health conditions and oral microbial data collected from the subject’s saliva as possible explanatory variables. The second model was the same as the first model except that the response was *Candida*-positive in plaque at each visit and the microbiome data were collected from the subject’s plaque. Subjects’ information from all three visits was used to fit the models. The final subsets of selected variables for both models were the same, resulting in overall identical models, so only one is reported here (Table 1). The coefficients for this model can be interpreted as follows: a positive sign indicates that the presence or increase of that variable increased the odds of *Candida* detection. A negative sign indicates that the presence or increase of that variable decreased the odds of *Candida* detection. For example, the coefficient for IP-10 (or CXCL10) was 0.399. As this was a model with a binary response, a logit-link was used, so to directly interpret the coefficient we must exponentiate it, exp (0.399) = 1.49. Therefore, we can say that the odds of detecting *Candida* were increased by a multiplicative factor of 1.49 for a one unit increase in IP-10 (or CXCL10) detection.

The second set of models was generalized linear models with a LASSO penalty (no mixture component). The response in these models was whether or not the subject responded to the Nystatin rinse, i.e., whether the *Candida* level decreased to an undetectable level from baseline to the second visit. Since our goal was to build models to predict the response to Nystatin using the baseline information, the baseline explanatory variables (demographic characteristics, medical background, oral health conditions, and oral microbial data) were used to fit the models. The two models were again differentiated by their use of the salivary oral microbial data versus the plaque oral microbial data. Different variables were selected in these models, and as such, both are reported here (Table 1). The interpretations for these models are similar mathematically, but the signs have the opposite meaning. A negative sign indicates that the variable in question had a detrimental effect on the response to Nystatin, whereas a positive variable indicates that the variable in question increased the odds of responding to Nystatin. Per the plaque model, we can again interpret the IP-10 (or CXCL10) coefficient (−0.180) by exponentiating it (exp (−0.180) = 0.835), and say that a one unit increase of IP-10 (or CXCL10) has a multiplicative effect of 0.835 on the odds of the subject responding to Nystatin. 

## 3. Results

### 3.1. Results

Details of the participants’ demographic, medical and oral health information have been reported previously [17]. Among the participants, 70% were female, 40% were White, 10% were Black and the remaining 50% reported with more than one race or with other racial background. Fifteen percent of the participants had hypertension, 20% reported smoking, and the average DMFT was 9.0 ± 5.8. Briefly, 19 participants completed the 1-week and 3-month follow-up visits. Among the total of 28 doses, the average number of medication doses used by participants was 24.4 ± 4.8. One participant withdrew due to reported side effects from using Nystatin oral rinse and did not complete the 7-day rinse and the 1-week and 3-month follow-up visits. Ten participants responded to Nystatin oral rinse without detectable salivary *C. albicans* at the 1-week follow-up visit, while the other nine participants remained *C. albicans* positive at the follow-up visits.

#### 3.1.1. Alpha and Beta Diversity Result

The alpha diversity of the oral microbiome in samples collected from participants who responded to the Nystatin rinse remained stable over the course of the study period. This is evident in Figure 1A,B, where the *p*-values were all greater than 0.05. Specifically, the *p*-values at week 1 were 0.39 and 0.76 for the saliva and plaque samples, respectively, and the samples collected at three months had *p*-values of 0.87 for both groups. 

For the participants who did not respond to the Nystatin rinse, the alpha diversity of the saliva and plaque samples at the species level showed no significant differences from the first to the second and third visits (Figure 1A,B, *p*-values are greater than 0.05). Similarly, the change in alpha diversity was more significant at the second visit compared to the third visit, similar to the findings in the samples obtained from participants who responded. Figure 1C reveals that there was nearly no change in the beta diversity of the samples, regardless of the participants’ response to the Nystatin rinse.

The rarefaction curves plateau as the sequencing depth reaches 2000 (Appendix A), which indicates that the diversity can be fully captured at this point. 

#### 3.1.2. Differentially Abundant Taxa in Plaque after Rinse

The results obtained from the ANCOM_BC model, as depicted in Figure 2A, demonstrate the presence of various differentially abundant species over the three visits, including *Veillonella parvula*, *Actinomyces* sp._*HMT 171*, *Capnocytophaga sputigena*, *Gemella haemolysans*, *Selenomonas artemidis*, *Streptococcus HMT 0 71*, *Streptococcus oralis*, *Streptococcus intermedius*, *Streptococcus parasanguinis I*, *Leptotrichia* sp._*HMT 225*, *Schaalia* sp._*HMT 180*, *Prevotella* sp._*HMT 300*, *Prevotella pleuitidis*, *Rothia mucilaginosa*, *Actiniomyces massiliensis* and *Streptococcus salivarius thermophilus*.

The linear discriminant analysis (LDA) effect size (LEfSe) method, as shown in Figure 2B, further supports these findings by indicating that *Actinomyces* sp._*HMT 171* was more abundant at the baseline compared to week 1. Additionally, *Prevotella* sp._*HMT 300* and *Eubacteriales* were found to be more abundant at week 1. It is noteworthy that both methods identify *Actinomyces* sp._*HMT 171* and *Prevotella* sp._*HMT 300* as species that showed a significant difference between the baseline and week 1. 

#### 3.1.3. Core Microbiome Analysis of Dental Plaque

From the baseline to week 1, the core taxa at the genus level, characterized by a prevalence above 20% and a relative abundance greater than 1%, were *Streptococcus* and *Actinomyces*. The core microbiome remained unchanged, as evidenced in Figure 3A,B. However, three months post-baseline, *Veillonella* emerged as a new core genus alongside *Streptococcus* and *Actinomyces* (Figure 3C).

#### 3.1.4. Network Analysis of Dental Plaque

The correlation network at the genus level was established through the application of Pearson’s correlation, which was utilized to calculate the pairwise correlations between taxa at the genus level in dental plaque samples collected from participants who responded to Nystatin oral rinse. The findings, depicted in Figure 4, indicate a multitude of correlations between microorganisms. Notably, a correlation was observed between *Eubacterium XIVaG 1* and *Prevotella*, which underwent a substantial change in relative abundance from the baseline to week 1.

#### 3.1.5. Relative Abundance of Species in Saliva and Plaque Samples

The relative abundance of species in saliva and plaque samples collected from subjects who responded to Nystatin rinse (Appendix A) and from those who did not respond to it (Appendix A) was analyzed. The bars in the figures are organized based on the time point of collection (baseline, 1 week and 3 months), and the top 38 most abundant species are presented. It can be observed in all four figures that the microbial composition of the samples underwent slight changes across visits in both saliva and plaque samples obtained from both responding and non-responding subjects.

### 3.2. Classification Analysis Results

While the classification models differed in their representation of the oral microbiome, the biomarker IP-10, also known as CXCL10, was selected for all models (Table 1), each time indicating a higher likelihood of *Candida*, or equivalently a poorer response to the Nystatin rinse. This would indicate that patients with higher levels of the biomarker IP-10 (or CXCL10) are not as strong candidates for the Nystatin rinse as those with lower levels.

## 4. Discussion

### 4.1. Oral Microbiota Changes following Nystatin Oral Application

Our previous study [16] found that Nystatin application could alter the formation and characteristics of *C. albicans-S. mutans* duo-species biofilms in vitro. However, further research is needed to ascertain the effect of Nystatin in microbiome communities. Even though no change in the alpha and beta diversity of the oral microbiome was observed in this study, we identified certain differentially abundant species after the participants were treated with the oral Nystatin rinse. The species found via the ANCOM_BC model to be significantly different between the baseline and the follow-up visits were *Veillonella parvula*, *Actinomyces* sp._*HMT 171*, *Capnocytophaga sputigena*, *Gemella haemolysans*, *Selenomonas artemidis*, *Streptococcus Oral Taxon 71*, *Streptococcus oralis*, *Streptococcus intermedius*, *Streptococcus parasanguinis I*, *Leptotrichia* sp._*HMT 225*, *Schaalia* sp._*HMT 180*, *Prevotella* sp._*HMT 300*, *Prevotella pleuitidis*, *Rothia mucilaginosa*, *Actiniomyces massiliensis* and *Streptococcus salivarius thermophilus*. Furthermore, another model LEfSe specifically indicated that *Actinomyces* sp._*HMT 171* was significantly more abundant at the baseline than at the 1-week follow-up. In contrast, *Prevotella* sp._*HMT 300* and *Eubacteriales* were in significantly higher abundance at week 1 than at the baseline. Among these species, several species are worth noticing and discussing in detail regarding their dental disease-causing ability. These taxa are *Streptococcus*, *Actinomyces*, *Prevotella* and *Veillonella.*

### 4.2. Efficacy of Nystatin Suspension in Treating Oral Candidiasis

The efficacy of Nystatin as an antifungal treatment for denture stomatitis and oral candidiasis was investigated in several studies. Their study results showed that Nystatin pastille was more effective in treating denture stomatitis compared to the placebo, while Nystatin suspension was inferior to other antifungal treatments such as miconazole, gentian violet and ketoconazole in treating oral candidiasis [27]. The duration of Nystatin treatment would also affect the clinical and mycological cure rates, as shown by several studies. The clinical and mycological cure rates were 9% to 63.5% and 6% to 13%, respectively, when Nystatin suspension was given for two weeks [28,29,30,31]. A higher cure rate 16.7% was reached if Nystatin suspension was applied for three weeks. The clinical and mycological cure rates of studies in which Nystatin pastilles was used for two weeks were 14.3% to 28.6% and 57.1% to 71.4% [32]. Moreover, treating the participants with Nystatin pastille for four weeks could raise the clinical cure rate to 76.9% and the mycological cure rate to 40% [33]. These all suggest that administering Nystatin pastille for a longer duration could raise the cure rates. 

In our study, participants were treated with Nystatin suspension and instructed to rinse their mouth with the solution four times a day for one week. However, only half of the participants were free of oral *C. albicans* detection at the follow-up visit. The microbiome results showed that there was no significant change in the alpha and beta diversity of the oral microbiota after the Nystatin suspension treatment. However, considering the higher cure rates observed when Nystatin pastille was used and when the treatment was administered for a longer period, future studies could attempt to use Nystatin pastille alone or in combination with Nystatin suspension for a longer duration to investigate the potential impact on the oral microbiota.

### 4.3. Association between IP-10 Level and Nystatin Response

Interferon (IFN)-γ-inducible protein, CXCL10/IP-10, is a member of the CXC chemokine family with pro-inflammatory and anti-angiogenic properties, and plays a role in recruiting immune cells, particularly T cells, to sites of inflammation or infection [34]. Elevated levels of IP10 in humans can be observed in various conditions, including viral infections, autoimmune disorders, inflammatory diseases and certain types of cancer [35,36,37]. For example, IP10 levels can increase during viral infections such as hepatitis C [38], hepatitis B [39], human immunodeficiency virus (HIV) [40], influenza [41] and respiratory syncytial virus (RSV) [42]. Elevated IP10 levels have been observed in autoimmune diseases such as rheumatoid arthritis [43], systemic lupus erythematosus (SLE) [44], systemic sclerosis [45] and multiple sclerosis [46]. In our study, it was found that a higher level of IP-10 (or CXCL10) was correlated with a poorer Nystatin response. This indicates that IP-10 (or CXCL10) is a predictive factor of Nystatin response, potentially because an elevated IP-10 reflected a compromised immune system of the individuals. 

### 4.4. Altered Relative Abundance of Actinomyces and Veillonella Suggest Anti-Caries Potential of Nystatin

A previous study [47] revealed that propionic acid, which is a product of carbohydrate metabolism, could also lead to a decrease in the pH level, resulting in demineralization of enamel hydroxyapatite crystals and proteolytic degradation of tooth hard tissue structures. The study established the involvement of *Actinomyces* in this process. Furthermore, a study conducted by Junko Kawashima [48] highlighted the cariogenic potential of *Actinomyces*, owing to its ability to produce acid, which is more tolerant to fluoride treatment than that of *Streptococcus*. However, our study found that the relative abundance of the genus *Actinomyces* decreased at the 1-week follow-up, compared to the baseline, as in both Figure 2A,B. Another study identified *Veillonella* species’ role in the conversion of lactic acid to weaker acids, such as propionic acid, which would increase the pH level [49]. Our study found that the relative abundance of *Veillonella parvula* increased after the use of Nystatin, as shown by Figure 2A. These findings altogether suggest that Nystatin acts in a way that can slow down the progression of demineralization of enamel hydroxyapatite crystals and proteolytic degradation of tooth hard tissue structures by affecting related oral micro-bacteria. 

### 4.5. The Decrease in Relative Abundance of Streptococcus Suggests Potential of Nystatin to Treat Recurrent Aphthous Stomatitis (RAS)

*Streptococcus* was identified as strongly correlated with the recovery and progression of the recurrent aphthous stomatitis (RAS), especially in middle-aged patients [50]. In our study, *Streptococcus HMT 0 71*, *Streptococcus oralis*, *Streptococcus intermedius*, *Streptococcus parasanguinis I* and *Streptococcus salivarius thermophilus* were all found to be differentially abundant between the baseline and 1-week follow-up based on the ANCOM-BC model (Figure 2A). This indicates that *Streptococcus* species can be influenced by Nystatin oral rinse. Thus, Nystatin rinse could be considered as a potential treatment for RAS where an individual’s progression and recovery is highly associated with *Streptococcus* [50].

### 4.6. Limitation

We recognized that with the limited sample size, the results generated from the study could not be applied to a general population. We did not conduct longer-term analysis to determine whether these taxa abundance changes are sustained and whether any potential return to baseline are reflected by return of *Candida*. 

## 5. Conclusions

Despite 1-week Nystatin oral rinse only being successful in eliminating oral *C. albicans* among nearly half of the participants in the clinical trial, the rinse was associated with specific oral taxa abundance change, such as that of *Streptococcus*, *Actinomyces*, *Prevotella* and *Veillonella*. Intriguingly, an elevated salivary Interferon (IFN)-γ-inducible protein (IP-10) level was a predictive factor of poor Nystatin response following Nystatin oral rinse. Future clinical trials are warranted to comprehensively assess the impact of antifungal treatment on the oral flora. 

## Figures and Tables

**Figure 1 microorganisms-11-01497-f001:**
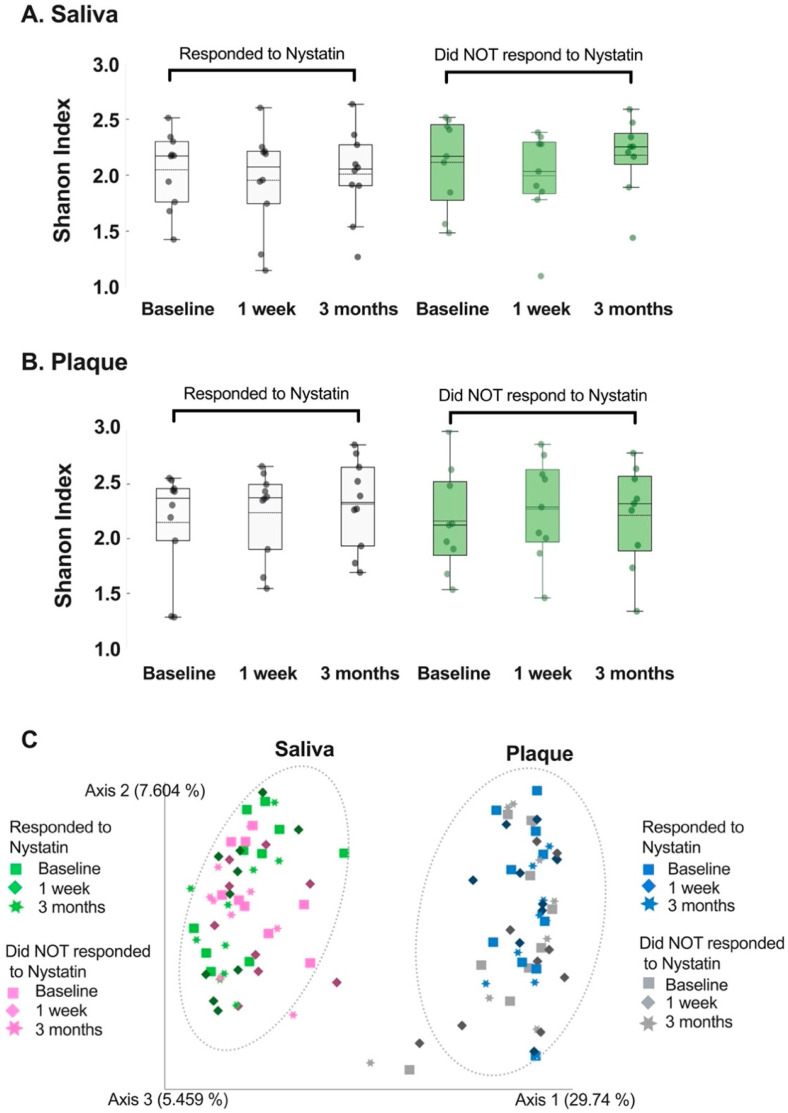
Community diversity of salivary and dental plaque microbiome. Alpha diversity of the salivary microbiome (**A**) and plaque microbiome (**B**) measured via the Shannon index. The participants are stratified based on their response to Nystatin oral rinse. Responding to Nystatin is defined as not detecting *Candida* at the follow-up visits. (**C**) Beta diversity was measured via the Bray–Curtis index. A principal coordinate analysis (PCOA) plot is generated. Permutational MANOVA (PERMANOVA) was used for these statistical comparisons between groups. No statistical significance between visits was identified for either alpha or beta diversity.

**Figure 2 microorganisms-11-01497-f002:**
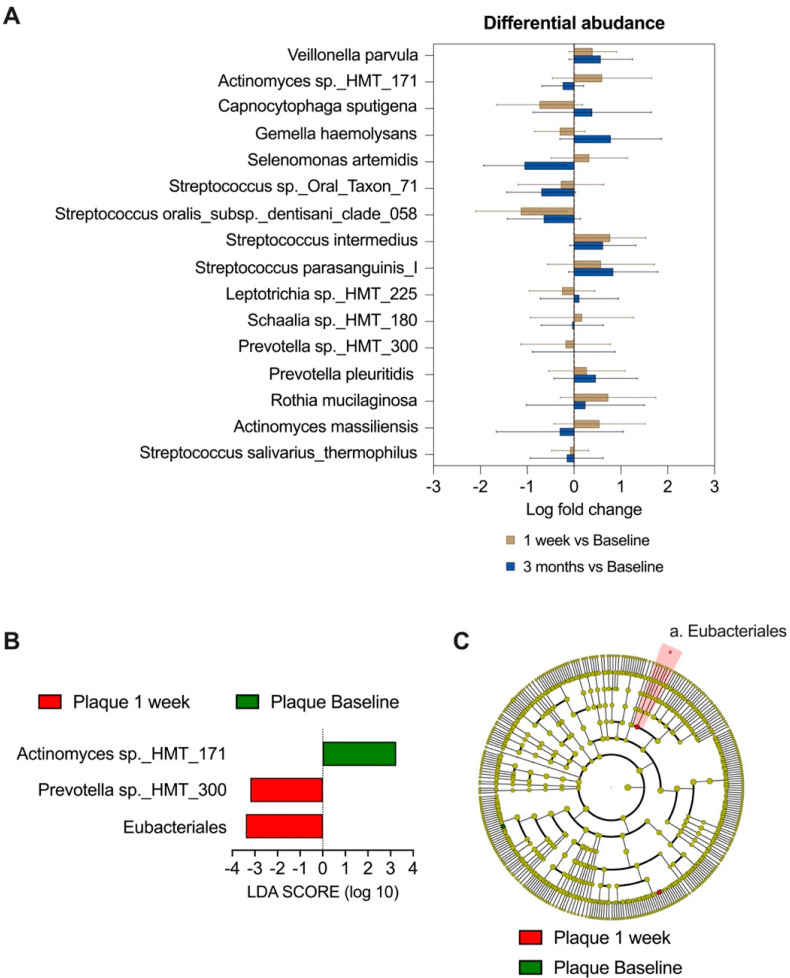
Differential Abundance Features in Dental Plaque Following Nystatin Oral Rinse. (**A**) Taxa at species with different abundance in dental plaque at different study visits. Data are represented by mean and 95% confidence interval bars (two-sided: Bonferroni adjusted) derived from the ANCOM-BC model. All species listed had significant differences with *p* < 0.05. (**B**) Taxa differently enriched in dental plaque at baseline and 1-week follow-up visits. The linear discriminant analysis (LDA) effect size (LEfSe) method was used to compare the taxa between the baseline and follow-up visits. The bar plot lists the significantly differential taxa based on effect size (LDA score log10 > 2.0 and FDR < 0.1). (**C**) Cladogram generated by LEfSe indicating differences in taxa between the baseline and 1-week visit. The green dot indicates the plaque at the baseline visit was enriched with *Actinomyces* sp._*HMT_171*. The red dot indicates that the plaque at the 1-week follow-up visit was enriched with *Prevotella* sp._*HMT_130* and *Eubacteriales*.

**Figure 3 microorganisms-11-01497-f003:**
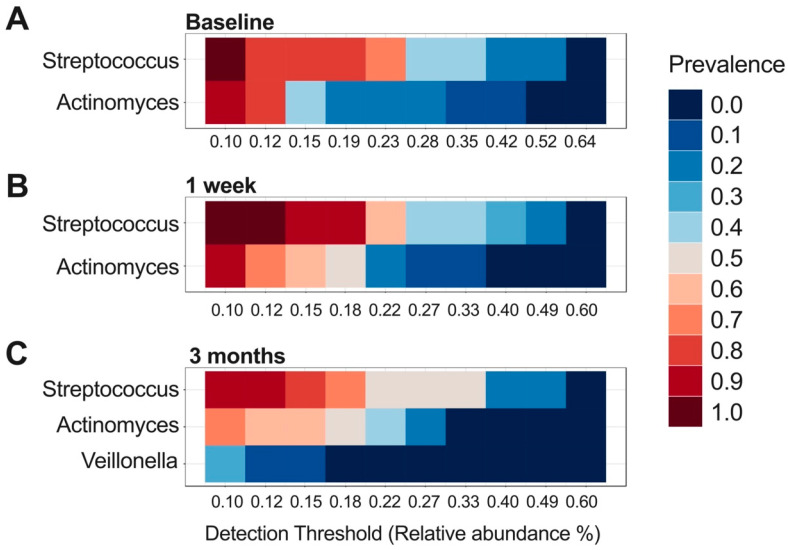
Core Genera in Dental Plaque among Participants Who Responded to Nystatin Oral Rinse. The core microbiome of dental plaque among participants who responded positively to Nystatin oral rinse (defined as no detection of *Candida* species at the follow-up visits). Taxa at the genus level with more than 20% prevalence and more than 1% relative abundance are depicted in (**A**) baseline, (**B**) 1-week follow-up and (**C**) 3-month follow-up visits.

**Figure 4 microorganisms-11-01497-f004:**
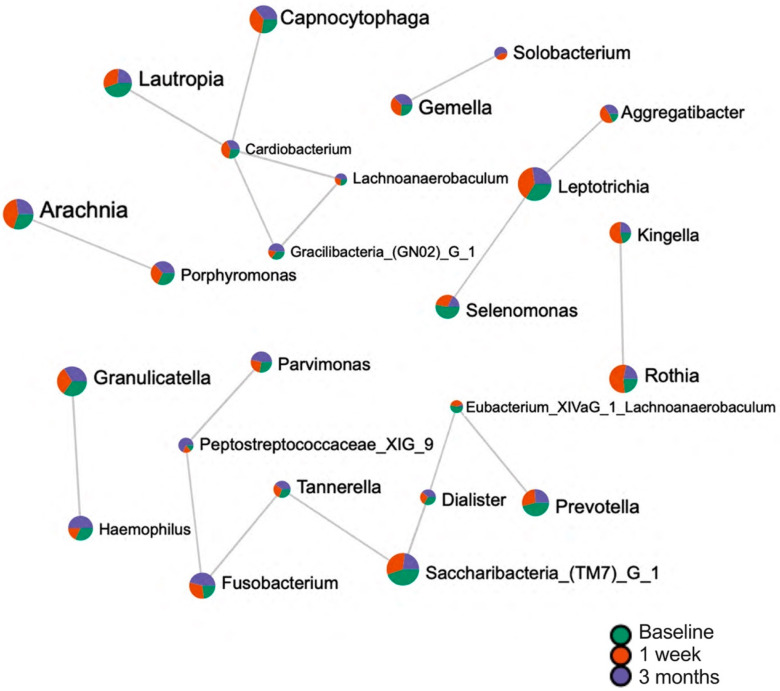
Network Correlation in Dental Plaque among Participants Who Responded to Nystatin Oral Rinse. A correlation network was built using Pearson’s correlation to calculate pairwise correlations between taxa at the genus level among dental plaque whose participants responded to Nystatin oral rinse. The correlation filter was set up as >0.6 and *p* < 0.05. The correlation results identify potential interactions between microbes that could represent mutualistic, commensal or competitive relationships. The pie chart indicates the mean of taxa-relative abundance at the related study visit.

**Table 1 microorganisms-11-01497-t001:** Estimated coefficients for classification models.

Variable	GLMM-LASSO Model	GLM-LASSO Salivary Model	GLM-LASSO Plaque Model
**IP-10**	0.399	−0.181	−0.180
**Employed (Yes/No)**	−0.132	Not Selected	Not Selected
**Weighted Sweet Index**	0.360	Not Selected	Not Selected
**Eotaxin**	0.609	Not Selected	Not Selected
**MDC**	3.521	Not Selected	Not Selected
**IL-15**	−0.617	Not Selected	Not Selected
**IL-10**	0.604	Not Selected	Not Selected
** *Streptococcus oralis* **	Not Selected	Not Selected	−0.024
** *Campylobacter concisus* **	Not Selected	−0.075	Not Selected
** *Solobacterium moorei* **	Not Selected	−0.225	Not Selected
** *Peptoniphilaceae parvimonas* **	Not Selected	−0.114	Not Selected
** *Clostridiales peptostreptococcaceae* **	Not Selected	−0.030	Not Selected

## Data Availability

The data presented in this study are included in the manuscript and Appendix A. The sequences have been submitted to the NCBI Bioproject database under accession number Biosamples (SUB13063146) and SRA (SUB13063094). The project will be released on 31 December 2023 or upon the publication of the manuscript, whichever comes sooner.

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
