# Peer review of "Impact of Nystatin Oral Rinse on Salivary and Supragingival Microbial Community among Adults with Oral Candidiasis"

_microorganisms, 2023, doi:10.3390/microorganisms11061497_

Round 1

Reviewer 1 Report

The paper of  Lanxin Zhang al to be rejected:

According to the authors themselves, the small number of samples does not make the paper significant and conclusive

Reviewer 2 Report

Dear author(s)

Thanks for your good work and presentation.

However, some comments and suggestions could be considered during the manuscript’s revision:

  1. Line 15, the statement “Our previous study indicated that Nystatin antifungal treatment could reduce the oral 15 carriage of Streptococcus mutans associated with dental caries” is not suitable here. It could be included in the introduction or discussion section and supported with references.
  2. Line 17, how can we describe healthy person with oral candidiasis?!
  3. Line 19, “Trials.gov #NCT04550546” should be included in the materials and methods section, not here.
  4. Line 26, the abbreviations “IP-10, also known as CXCL10” should be mentioned as full words. All abbreviations in the whole manuscript should be mentioned.
  5. The aim of this study should be mentioned at the end of the abstract section.
  6. The name of “nystatin” should be written in the same way.
  7. Lines 43 & 57, the scientific name of the fungus should be revised. The scientific name of fungi should be written as full names at the first mention, then mentioned as an abbreviation (). This abbreviation should be mentioned in the whole manuscript.
  8. Line 46, the same previous comment for the bacterial spp.
  9. Do you consider mouth rinsing “four times a day (every 6hours)”for a week is a practical method?!
  10. Do you think that twenty participants are enough for evaluation?
  11. In which basis there was significant differences among groups (statistical differences)?
  12. A more focusing on the elevated level of IP-10 should be addressed in the discussion. 

Best wishes  

Just minor revision.

Reviewer 3 Report

In this study, the authors investigated the effects of nystatin oral rinse on salivary and supragingival microbial community among healthy adults. The study was well designed and conducted, analysis was correctly performed, and the results were carefully discussed. The manuscript could be accepted after minor revision:

1)      The authors might briefly introduce Nystatin in the introduction.

2)      There should be consistency throughout the manuscript using past tense when quoting another person's research.

3)      Use the same unit format according to the journal requirements. For example, don’t use ml (line 124) and mL (line 98) at the same time.

4)      Line 132, “Candida spp.” should be italics. And please carefully check the whole manuscript to revise the similar mistakes.

5)      Check and revise the formation of the references according to the journal requirements.

The manuscript was well written, however there are still some English errors need to be revised.
